# Cytosolic EpCAM cooperates with H-Ras to regulate epithelial to mesenchymal transition through ZEB1

Fatma A. Omar[1], Taylor C. Brown[2], William E. Gillanders[2], Timothy P. Fleming[1], Michael A. Smith[1], Ross M. Bremner[1], Narendra V. Sankpal [1]*

1 Norton Thoracic Institute, St. Joseph's Hospital and Medical Center, Phoenix, Arizona, United States of America, 2 Siteman Cancer Center, Washington University School of Medicine, St. Louis, Missouri, United States of America

* narendra.sankpal@commonspirit.org

**Editor:** Erika Di Zazzo, University of Molise Department of Medicine and Health Sciences "Vincenzo Tiberio": Universita degli Studi del Molise Dipartimento di Medicina e Scienze della Salute Vincenzo Tiberio, ITALY

## Abstract

Next generation sequencing of human cancer mutations has identified novel therapeutic targets. Activating Ras oncogene mutations play a central role in oncogenesis, and Ras-driven tumorigenesis upregulates an array of genes and signaling cascades that can transform normal cells into tumor cells. In this study, we investigated the role of altered localization of epithelial cell adhesion molecule (EpCAM) in Ras-expressing cells. Analysis of microarray data demonstrated that Ras expression induced EpCAM expression in normal breast epithelial cells. Fluorescent and confocal microscopy showed that H-Ras mediated transformation also promoted epithelial-to-mesenchymal transition (EMT) together with EpCAM. To consistently localize EpCAM in the cytosol, we generated a cancer-associated EpCAM mutant (EpCAM-L240A) that is retained in the cytosol compartment. Normal MCF-10A cells were transduced with H-Ras together with EpCAM wild-type (WT) or EpCAM-L240A. WT-EpCAM marginally effected invasion, proliferation, and soft agar growth. EpCAM-L240A, however, markedly altered cells and transformed to mesenchymal phenotype. Ras-EpCAM-L240A expression also promoted expression of EMT factors FRA1, ZEB1 with inflammatory cytokines IL-6, IL-8, and IL1. This altered morphology was reversed using MEK-specific inhibitors and to some extent JNK inhibition. Furthermore, these transformed cells were sensitized to apoptosis using paclitaxel and quercetin, but not other therapies. For the first time, we have demonstrated that EpCAM mutations can cooperate with H-Ras and promote EMT. Collectively, our results highlight future therapeutic opportunities in EpCAM and Ras mutated cancers.

## Introduction

Epithelial cell adhesion molecule (EpCAM), a homophilic transmembrane adhesion protein, is a highly studied tumor antigen. EpCAM is overexpressed in most epithelial tumors and is frequently used as a marker to identify circulating tumor cells and cancer stem cells. As such,

**Data Availability Statement:** All relevant data are within the paper and its Supporting Information files.

**Funding:** This research was supported by a startup grant from Norton Thoracic Institute, St. Joseph's Hospital and Medical Center to NS. NO. The funders had no role in study design, data collection and analysis, decision to publish, or preparation of the manuscript.

**Competing interests:** The authors have declared that no competing interests exist.

EpCAM has been a frequent target of monoclonal antibody therapy [1, 2]. However, EpCAM antibody therapies have shown only partial success [2–4]. It has been predicted that altered EpCAM localization and protease-mediated cleavage may inhibit the effectiveness of antibody therapy. Recently, we identified 115 unique cancer-associated somatic/missense mutations in the EpCAM coding region [5]. Our database study shows up to 5% of the lung cancer cases harbored EpCAM mutations and more than half of these mutants were exclusively expressed in the cytosolic compartments.

In addition to its role as an adhesion molecule, EpCAM can function as a signaling molecule and promote or suppress tumor progression, depending on the cancer context [6]. Our previous work has shown that overexpressed EpCAM can alter signaling pathways involving NF-kB [7], JNK [8], and ERK1/2 [9]. Importantly, we demonstrated that wild-type (WT) EpCAM inhibits ERK1/2 signaling and suppresses epithelial-to-mesenchymal transition (EMT), invasion, and metastasis [5]. We have also demonstrated that the thyroglobulin type-I domain of WT EpCAM binds and inhibits cathepsin-L protease, which hinders cancer cell invasion and lung metastasis. When mutated, approximately half of all cancer-associated EpCAM mutations localize to the cytosolic compartments, where it cannot inhibit extracellular cathepsin-L [5].

The Ras family of proto-oncogenes is one of the most frequently mutated genes in cancer and has been intensely investigated as a potential therapeutic target. The Ras gene family encodes three small G proteins: H, N, and K-Ras [10]. Ras proteins are anchored on the cytoplasmic side of the cell membrane and mediate signal transduction downstream of a variety of effector molecules, which in turn activate a cascade of pathways, including RAF/MEK/MAPK, PI3K/AKT, and RAL–GDS. These 3 effector pathways play major roles in mediating signals related to cell proliferation, survival, adhesion, and cell motility. These roles are initiated and maintained by overexpression of a series of proteins by activated nuclear transcription factors [11–13]. Based on our previous observation in Ras-transformed cells, we hypothesized that EpCAM contributes to the oncogenic function of Ras genes. In the present study, we explored the role of EpCAM in conjugation with H-Ras. For the first time, we demonstrate that Ras expression increased EpCAM transcription and partially promoted cytosolic expression of EpCAM. EpCAM-L240A, a cancer- associated EpCAM mutant that is localized in the cytosol, altered normal epithelial cells to a mesenchymal phenotype through ZEB1 and cytokine signaling.

## Materials and methods

### Cell culture and reagents

MCF-10A cell lines were obtained from the American Type Culture Collection (ATCC, Rockville, MD, USA). MCF-10A cells were grown in Mammary Epithelial Cell Growth Medium (Lonza, Allendale, NJ, USA) supplemented with 5% horse serum. Recombinant epidermal growth factor (EGF) and transforming growth factor-β (TGFβ) were purchased from R&D Systems (Minneapolis, MN, USA).

The MAPK inhibitors CI-1040, SB203580, and SP600125 and AKT inhibitor LY29004 were purchased from Selleck Chemicals (Houston, TX, USA). BAY117082, dasatinib, doxorubicin, etoposide, paclitaxol and quercitin were purchased from LC labs (Woburn, MA, USA).

### Constructs

The full-length EpCAM cDNA was amplified from the MCF-10A mammary epithelial cell line and sub-cloned into pBabe retroviral vector and pLL3.7 lentiviral vector as described earlier [5]. The nucleotide sequence was confirmed with NCBI reference sequence NM_002354.

pBabe H-Ras was procured from Addgene (Watertown, MA, USA; cat#18744). pBabe-Puro was a kind donation from Dr. Sheila Stewart (Washington University). Other plasmids used were procured from Addgene; C-myc (#16011), Src (#13663), B-Catenin (#13371), E2F3 (#37970) and MEK1 (#53195).

### Retroviral and lentiviral transduction

Phoenix-AMPHO (ATCC) packaging cells ($2x10^6$) were transfected when nearly confluent with 2.5 μg of pBABE-H-Ras-Puro using FuGENE-HD (Promega, Madison, WI, USA). Twenty-four hours after transfection, the medium was replaced with 10% fetal bovine serum. Forty-eight hours after transfection, viral supernatants were collected, filtered through 0.45-micron filters, and then added to MCF-10A cells in media containing 8 μg/ml protamine sulfate. After two successive retroviral infections, cells were grown for 48 hours and selected in puromycin for 2 weeks. EpCAM-EGFP fused DNA was transiently transfected into HEK-293 T cells using FuGENE6 or stably transduced as described earlier [5].

### Immunofluorescence and cell staining

For immunofluorescence assay $2X10^5$ cells were grown on 8-well microscope slides. Following growth factor treatment, cells were washed twice with phosphate-buffered saline (PBS) and fixed at room temperature using 4% formaldehyde. Following fixation, cells were washed four times with PBS, permeabilized for 10 min at room temperature with 0.2% triton X-100 in PBS (v/v), washed 4 additional times and then blocked with 5% goat serum for 1 h. After blocking, cells were incubated with primary antibody overnight at 4°C or 2 h at room temperature in PBS containing 5 mg/ml bovine serum albumin. Cells were washed 5 times and incubated with appropriate secondary antibody conjugated with Alexa-flour 488 or 555 (Invitrogen, Waltham, MA, USA) for 1 h at room temperature. Nuclear staining was carried out using 4,6-dia-midino-2- phenylindole (Invitrogen) for 5 min. The cells were then dehydrated and mounted and visualized/captured with a fluorescence microscope (EVOS digital inverted microscope at 20× or 40× magnification).

### Flow cytometry

Cell surface EpCAM expression levels were measured by flow cytometry using FITC or phyco-erythrin-labeled EpCAM antibodies. Briefly, $1X10^6$ cells were stained with EpCAM-PE antibody as recommended by manufacturer. Expression was quantified as mean fluorescence intensity (MFI) using a FACScan flow cytometer (BD Biosciences, San Jose, CA, USA).

### Confocal microscopy

Fixed or living cells ($2X10^5$ cells per assay) were plated and grown on coverslips. imaged on a confocal laser scanning microscope TCS SP5 (Leica Microsystems, Wetzlar, Germany) using a 63× oil immersion objective. A helium/neon (λ543 nm) laser was used for excitation of Rhoda-mine. Living cells were imaged in imaging medium (Dulbecco's modified Eagle's medium without phenyl red). For fixation, cells were washed in PBS once, fixed in 4% paraformalde-hyde for 10 min on ice, and then returned to PBS for imaging. To quantify rhodamine-EGF signal, the sum of pixel values was calculated from the raw images. For this, an ImageJ plugin was written that is available upon request. For normalization and comparability with flow cytometry data, the average pixel value obtained at each time point was divided by the maximal signal obtained at 30 min of stimulation.

## Western blotting

From 6-cm culture plate, sub-confluent cells were washed with ice cold PBS and lysed in RIPA cell lysis buffer with a protease inhibitor cocktail (Cell Signaling Technology, Danvers, MA, USA). Protein concentrations were determined by bicinchoninic acid protein assay (Pierce, Rockford, IL, USA). In all assays, 20–30 μg of protein was subjected to sodium dodecyl sulfate-polyacrylamide gel electrophoresis (NuPAGE, Life Technologies, Carlsbad, CA, USA) and transferred by electrophoresis to a polyvinylidene difluoride (PVDF) membrane and developed with following antibodies; EpCAM (SCBT,sc-25308), Actin-HRP (SCBT ,sc-47778), p-ERK1-T202,Y204 (Cell Signaling #4370), ERK1/2, (Cell Signaling #7102), pJNK-T183,Y185 (Cell Signaling #4668), JNK, (Cell Signaling #9258), pP38-T180/Y182 (Cell Signaling #4511), P38 (Cell Signaling #9212), pATK-S473 (Cell Signaling #4051), AKT, (Cell Signaling #9212), pP65-S536 (Cell Signaling #3033), P65 (Cell Signaling #8242), pSrc-Y416 (Cell Signaling #2101), SRC, (Cell Signaling #2108), Cleaved PARP, (Cell Signaling #5625), Cleaved Caspase-3 (Cell Signaling #9661), Pan-Ras (SCBT, sc-166691), H-Ras (Cell Signaling #3339), E-cadherin (SCBT, Sc-7870), ZEB1 (SCBT, Sc-25388), FRA1 (Cell Signaling #5281), Vimentin (Cell Signaling #3516), Snail (Cell Signaling #3879), CTNNB1 Cell Signaling #8480). Signal detection was performed using the Super-Signal West Pico chemiluminescent immunodetection system (Thermo Scientific, Rockford, IL, USA). To quantify band density, immunoblots were developed on film and scanned, and pixels in each band were measured using the Image J software.

## RNA extraction and real-time PCR

For total RNA extraction, $2x10^6$ cells were plated for 24 h and purified using RNAeasy (Qiagen, Valencia, CA, USA). Two micrograms of RNA were reverse transcribed using a High Capacity cDNA Synthesis Kit (Ambion, Austin, TX, USA). mRNA expression was quantified using SYBR green chemistry and an ABI Prism 7700 Sequence Detector (Life Technologies). Primer sequences are detailed in **S1 Table**. Each reaction was performed in triplicate, and the data represents from 2 independent RNA preparations.

## Cytokine array

Cytokine antibody array was performed with a human cytokine array kit (R&D Systems, Minneapolis, MN, USA) according to the manufacturer's protocol. Briefly, 2-mL conditioned media in Opti-Mem from $2x10^6$ cell culture was collected after 72h. After centrifugation, the membranes pre-coated with capture antibodies were incubated with supernatants. After washing with wash buffer detection antibody was added followed by streptavidin-HRP. Chemi Reagent Mix were added to the membranes to develop the array blots. The immunoblot images were captured and visualized using the BioSpectrum Imaging System (Ultra-Violet Products Ltd., Cambridge, UK). The intensity of each spot in the captured images was analyzed using the ImageQuant 5.0 software (Molecular Dynamics, Sunnyvale, CA, USA).

## Cell proliferation and viability assay

For the cell proliferation and viability assay, $5.0 \times 10^3$ cells per well were plated in 96-well plates and cultured in a 5% $CO_2$ incubator. After 24 h, compounds were added as IC50 or concentrations reported in literature. Compounds were dissolved in DMSO and added to the culture, CI-1040 (20nM), SB203580 (100nM), SP600125 (1uM), AKT inhibitor LY29004 (2uM), BAY117082 (10uM), Dasatinib (100nM), doxorubicin (2uM), etoposide (5uM), paclitaxol (100nM), quercetin (5uM). After 72 to 96 h later, cell proliferation was assessed using the Cell Titer-Glo® Luminescent assay (Promega) following the protocols provided by the

manufacturers. Cell viability was determined using MTT assay (Millipore Sigma, Burlington, MA, USA). The viable cells reduced the MTT to formazan through oxidoreductase enzymes. The insoluble formazan crystals were solubilized resulting in a colored solution which is quantified by measuring absorbance at 590 nm. The measured absorbance at OD 590 nm is proportional to the number of viable cells.

### Invasion, migration and soft agar assay

Stably transduced $4 \times 10^4$ cells were added to matrigel transwell invasion chambers or control transwell chambers (BD Biosciences) and incubated for 72 h with chemoattractant media (Clonetics, Walkersville, MD, USA) supplemented with growth factors. Cells invading through the matrigel or control membranes were fixed using 70% ethanol, stained with 0.1% crystal violet, and photographed in 4 fields to cover the entire area. Cells were counted from all fields by a scientist blinded to the experimental conditions. For colony formation assay, $1.0 \times 10^4$ cells were plated into 6-cm dishes in 2 ml of medium containing 0.3% agarose, overlaid with 2 ml of 0.5% agarose. Dulbecco's Modified Eagle's Medium (Thermo Fisher Scientific, Waltham, MA, USA) was added on top, and dishes were maintained in a humidified atmosphere with 5% $CO_2$ at 37˚C for 12 days. Then colonies were stained with crystal violet, and the number of colonies was counted.

### Microarray data analysis

Gene expression profiles of 51 cancer cell lines (GSE12777), NCI-60 cell lines (GSE5846) and MCF10A-H-Ras cells (GSE3151, GSE12764,) were downloaded from the Gene Expression Omnibus data repository. CCLE breast cancer cell line data was downloaded from https://sites.broadinstitute.org/ccle/ used in S3 Fig. Unsupervised clustering of normalized data was used to generate heat plots using the TIBCO Spotfire Q6 software (Boston, MA, USA) and GENE-E (Broad Institute). A recently published gene set of 76 EMT signature genes was used to classify the cancer cell lines and primary cancers into epithelial- or mesenchymal-like cancers.

### Statistical analysis

All experiments were performed at least three times in triplicate. All statistical analyses were performed using GraphPad Prism 9.4 (GraphPad, La Jolla, CA). Numerical data are presented as mean ± sd. Single comparisons were performed by unpaired Student's t tests and multiple comparisons were performed by ANOVA. P-values <0.05 were considered to be statistically significant.

## Results

### H-Ras expression induces EpCAM expression

To establish the possible involvement of EpCAM in Ras signaling, we first analyzed genes correlated to E-cadherin. Microarray data from analysis of cancer cell lines revealed 121 genes, including EpCAM, closely correlated (R>0.6) to the bonafide epithelial marker E-cadherin [14]. Of these 121 epithelial genes, 35 genes were closely correlated (R>0.7) with EpCAM (Fig 1A). We subsequently analyzed expression of top 14 highly EpCAM correlated genes from Ras- transduced normal cells (Fig 1A, right panel). We extended our analysis to NCI-60 cell lines database with Ras mutations (S1B Fig). Data revealed that EpCAM correlated gene (nearest neighbors) list from Figs 1A and S1B are enriched for epithelial specific markers. Two independent microarray data sets were analyzed from immortalized MCF-

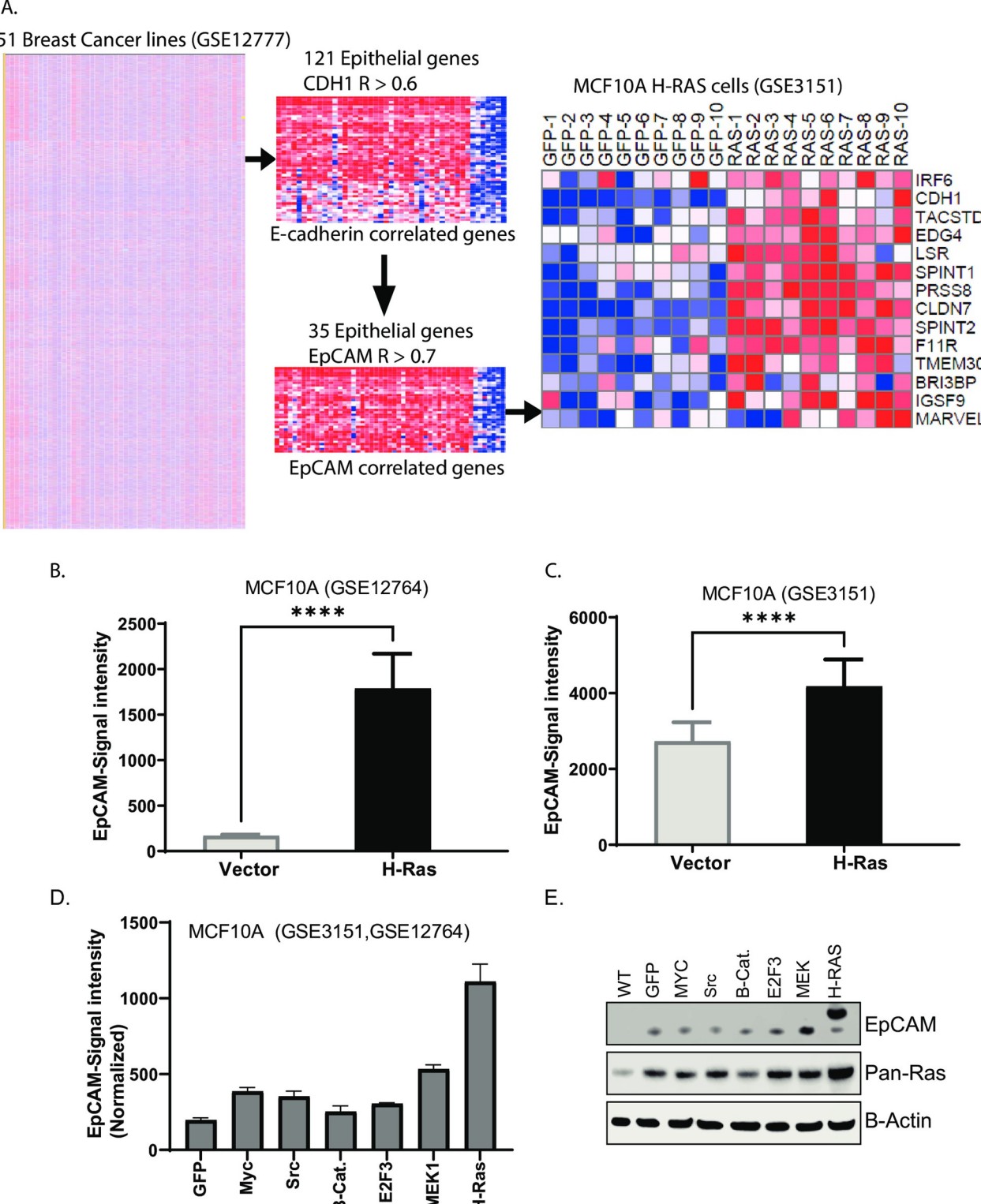

**Fig 1. Ras-induced EpCAM and several other membrane proteins.** A. Microarray data (GSE12777) for 51 breast cancer cell line was analyzed using Gene-E software. E-cadherin correlated 121 genes are enriched in first step followed by EpCAM close (correlated). These genes are presented in normal and Ras-transduced cells (right panel). B-C. Two independent microarray data sets from MCF10A and Ras-transduced cells were analyzed for EpCAM expression. D. Normalized Microarray data (GSE3151, GSE12764) shows compared to oncogenes myc, Src, B-catenin, E2F3 and MEK, Ras- transduced cells show EpCAM overexpression. E. Immunoblot assay after 96h of transiently transfected MCF10A cells with GFP, Myc, Src, B-Catenin, E2F3 MEK1 and H-Ras shows, H-Ras induced expression of EpCAM.

10A cells transduced with H-Ras [15, 16]. Gene expression data revealed that H-Ras-transduced cells upregulated EpCAM expression (Fig 1B and 1C). Further analysis and immunoblotting confirmed that compared to expression of a control vector or other proto-oncogenes, including myc, Src, β-catenin, E2F3, and MEK, Ras-transduced cells had a 5-fold increase in EpCAM expression (Fig 1D). H-Ras transduced cells validated this observation (Fig 1E) Altogether, this data suggests Ras overexpression induced expression of several epithelial/membrane proteins including EpCAM, which may play important role in Ras signaling. This is in agreement with a previous observation that K-Ras addiction is restricted in epithelial cells expressing EpCAM, but not in mesenchymal cells [17]. Based on previous studies, it is highly likely EpCAM plays a role in Ras-addicted cells.

## EpCAM and H-Ras transforms the epithelial cells

To establish a functional model to study the role of EpCAM in Ras cells, MCF-10A cells were transduced using H-Ras retrovirus followed by WT-EpCAM and EpCAM-L240A. We have repeatedly observed that EpCAM expression varies in a panel of epithelial tumor cell lines [9]. Normal, immortalized MCF-10A cells require EGF and insulin to proliferate. Growth factor supplementation suppresses EpCAM expression through ERK/SLUG signaling [9]. However, withdrawal of growth factor re-express EpCAM (S1A Fig). Throughout the studies, cells were cultured in the presence of growth factors (EGF and insulin) to keep endogenous EpCAM suppressed. When analyzed by flow cytometry, normal MCF-10A cells expressed endogenous EpCAM at very low levels (70–90 MFI). Moderate EpCAM expression in cancer cell lines varies from 400–600 MFI [9]. Overexpression ranges from 700–900 MFI. Retroviral Ras-transduced MCF-10A cells demonstrated EpCAM expression at 400–600 MFI when cultured without growth factors. With growth factors, endogenous EpCAM remained suppressed. On the other hand, EpCAM transduction increased the MFI to 600. Surface staining with EpCAM-PE antibody did not detect the mutant cytosolic EpCAM-L240A expression. (S1A Fig).

To generate Ras and EpCAM stable lines, MCF10A cells were transduced with retrovirus harboring control vector (pBabe) or H-Ras. After selection in puromycin, both cell lines were further transduced with pBabe, EpCAM-WT, or EpCAM-L240A. During our previous study [5], we generated several cancer-associated EpCAM mutations to test the effect of cytosolic EpCAM expression on cathepsin-L protease activity. Compared to other mutations, EpCAM-L240A was exclusively and widely localized to the cytosol and maintained its expression. After subsequent transduction and selection of a uniform population, all 6 established cell lines were passaged for 2 months prior to functional investigation.

Compared to the other cell lines, H-Ras-EpCAM-L240A-transduced cells showed a markedly altered phenotype (Figs 2A and S2). Since cells were grown in growth factor media, endogenous EpCAM expression remained marginal as expected (S1A Fig). H-Ras-EpCAM-L240A cells overgrew in colonies, clustered heavily together, and later released as spheroids in culture media. These spheroids, when re-plated, regrew as a monolayer with mesenchymal morphology. Thin sheet-like membrane protrusions consistent with lamellipodia formation were seen at the leading edge of the cells in H-Ras-pBabe and WT-EpCAM cells. In contrast, only filopodia were seen in H-Ras-EpCAM-L240A cells. Lamellipodia and filopodia appearing on the leading edges of cell movement are often observed in migratory cells. Tumor cells often evolve to become invasive and migratory allowing them to spread to distant sites.

Boyden chambers were used to investigate the migratory/invasive ability of the tested cell lines. Cells were plated onto inserts coated with a layer of extracellular matrix. After 72 h of invasion, cells were fixed, stained with crystal violet, and quantified. As shown in Fig 2B, H-Ras-transformed cells were partially invasive compared those transformed with control

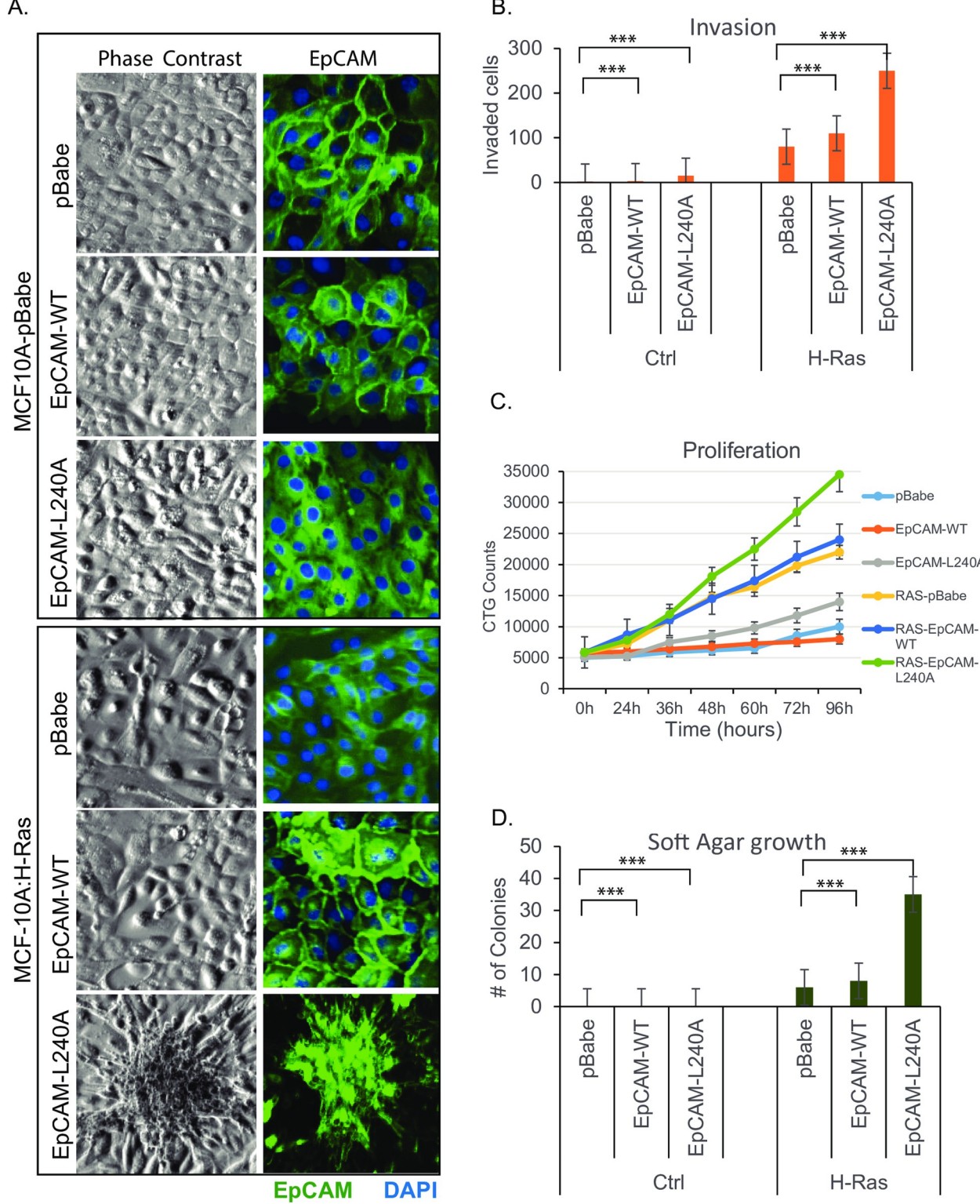

**Fig 2. Cytoplasmic EpCAM and Ras transformed normal MCF10A cells.** A. Cells were stably transduced with lentiviral GFP or EpCAM-GFP to monitor localization. Phase contrast images showing cell morphology after 24h of plating. B, Matrigel chamber invasion assay show Ras cells with EpCAM-L240A had higher levels of cell in invasion than controls. Matrigel invasion assay was performed for 72h. C. For cell proliferation analysis, plated cells were grown for 96h and assayed using the CellTiter-Glo Luminescent Cell Viability Assay. Data shows, Ras transduced cells proliferated more and compared to H-Ras, H-Ras-EpCAM-L240A proliferated at increased levels. Cells on soft agar was cultured for 12 days. D. Colony formation ability of cells was determined by soft-agar assay as described in methods. Compared to pBabe and EpCAM-WT, EpCAM-L240A cells formed more colonies suggesting ability to grow as Anchorage-independent growth and transformed phenotype.

vector alone. Compared to H-Ras control cells, EpCAM-WT and EpCAM-L240A cells showed a 2- and 3-fold increase in invasion, respectively. Additionally, H-Ras cells expressing EpCAM WT or L240A mutant showed increased proliferation (Fig 2C) in 72h assay. The tumorigenic potential of transformed cells was also investigated using soft-agar growth assays to confirm cellular anchorage-independent growth. As described in the methods section, cells were cultured on soft agar with complete growth medium. As shown in Fig 2D, Ras-expressing cells started forming colonies. There was not a statistical difference between the growth of control vector and EpCAM-WT cells; however, H-Ras-EpCAM-L240A cells showed a robust 4-fold increase in colony numbers.

## Cytoplasmic localization of EpCAM in Ras- and TGF1β1-induced EMT

Based on the previous findings, we hypothesized that the morphological, invasive, and growth properties observed in H-Ras-EpCAM-L240A cells could be the result of EMT (Fig 2B–2D). To test this hypothesis, cells were cultured on cover slips and immuno-stained for epithelial markers E-cadherin and β-catenin and mesenchymal markers vimentin and ZEB1. As shown in Fig 3A, compared to H-Ras-EpCAM-WT cells, H-Ras-EpCAM-L240A cells lost E-cadherin expression, with increased expression of mesenchymal markers vimentin and ZEB1.

β-catenin is a critical component of cadherin-based intercellular adhesions and also plays a central role in Wnt signaling. β-catenin can be located at the cell membrane, the cytoplasm, or the nucleus. At the cell membrane, it is bound to the cytoplasmic domain of type-I cadherins and regulates and maintains the structural organization of the cytoskeleton [18]. When released from cadherins, β-catenin participates in downstream signaling and cytoplasmic localization of β-catenin and can serve as a predictive marker for poor outcomes in cancer patients [19]. Compared to normal and H-Ras-transduced cells, H-Ras-EpCAM-L240A cells show strong staining of β-catenin in the cytosol. Interestingly, EpCAM-WT is localized at the membrane, but H-Ras-transduced cells show moderate EpCAM localization in the cytosolic compartments (Fig 3A).

It is known that Ras transduction in normal epithelial cells results in a partial EMT phenotype mediated by SNAI1/2 transcription factors [20]; however, ZEB1 expression is exclusively observed in mesenchymal cells. We have shown that *in vitro* cultured cells treated with TGF1β1 can induce EMT [9]. To investigate if EMT-transitioning cells express EpCAM in the cytosol, HaCAT cells (EpCAM MFI 500) were treated with TGF1β1 for 36 h, permeabilized, and stained with EpCAM antibody. As shown in Fig 3B, EMT-transitioning cells show EpCAM localization on the membrane and in the cytosol. Cytosolic localization of EpCAM in EMT transitioning cells may degrade or play a role in EMT initiation. Altogether, our results suggest that that EMT transition drives EpCAM expression to the cytosol and EpCAM-L240A-induced EMT in Ras cells.

## Ras and EpCAM induces EMT

In light of the cytosolic localization of EpCAM and the altered phenotype in H-Ras-EpCAM-L240A cells, we hypothesized that EMT transcription factors together with other cancer-associated factors may be playing an important role. The EMT stage of epithelial cells has been well characterized by gene expression signatures [9, 21]. Differential gene expression of a panel of epithelial and mesenchymal markers as well as MMP protease family members and cytokines involved in EMT was determined using qRT-PCR. As shown in Fig 4A, transduction with H-Ras-EpCAM-WT did not significantly alter the expression mesenchymal genes. H-Ras-EpCAM-L240A cells, however, showed decreased expression of epithelial markers and robust overexpression of Snail, Twist1, ZEB1/2, Fra1/2 and Foxc2. These transcription factors

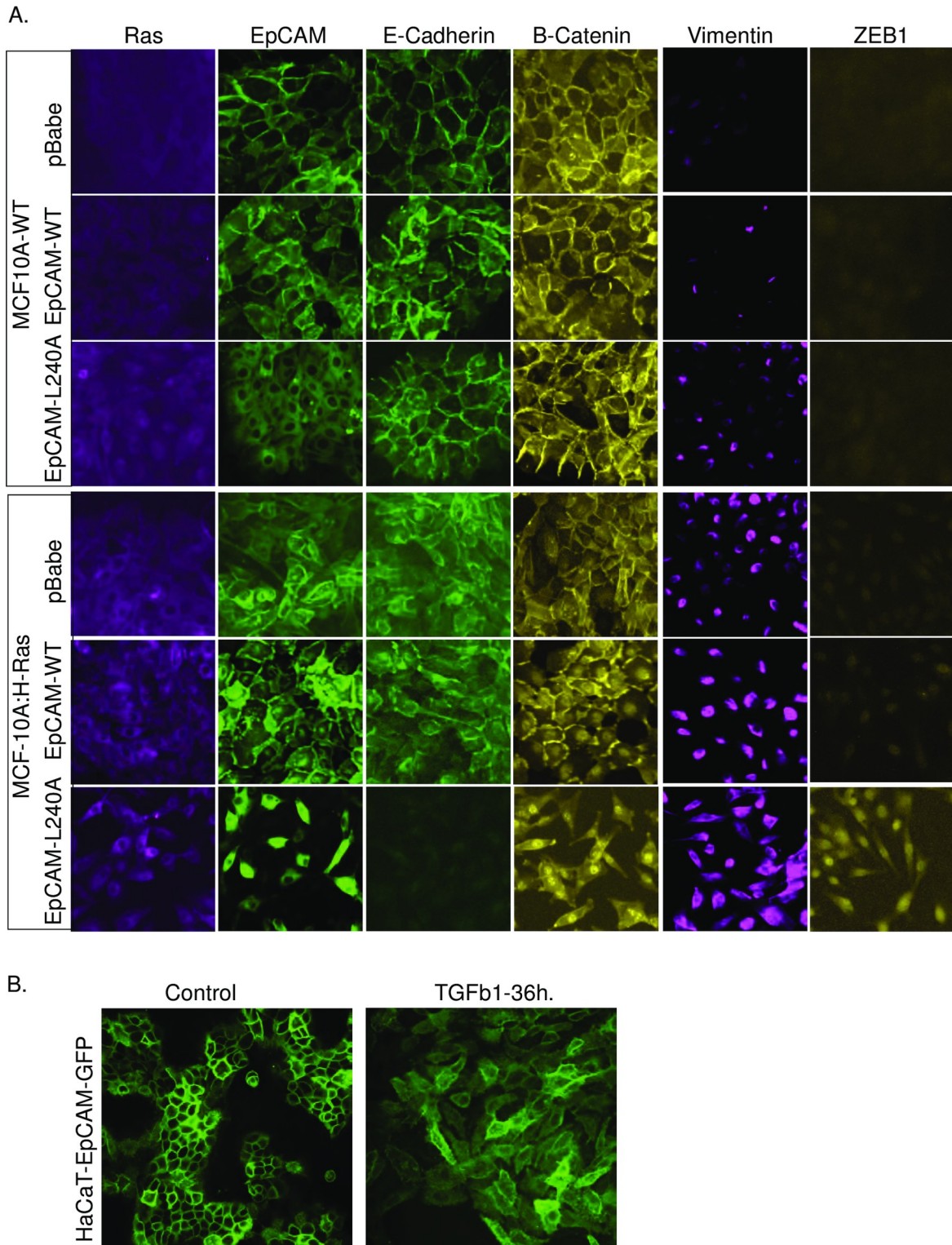

**Fig 3. Cytoplasmic EpCAM induced EMT.** A. MCF10A cells were transduced with control vector (pBabe) and Ras or EpCAM-WT and EpCAM-L240A. Cells were permeabilized, fixed, and stained with EpCAM, E-cadherin, β-catenin, vimentin and Zeb1 antibodies. Images were taken with EVOS Imaging System fluorescent microscope. B. HaCAT cells were treated with TGFb1 for 36 h and stained with EpCAM antibodies. Confocal images confirm localization.

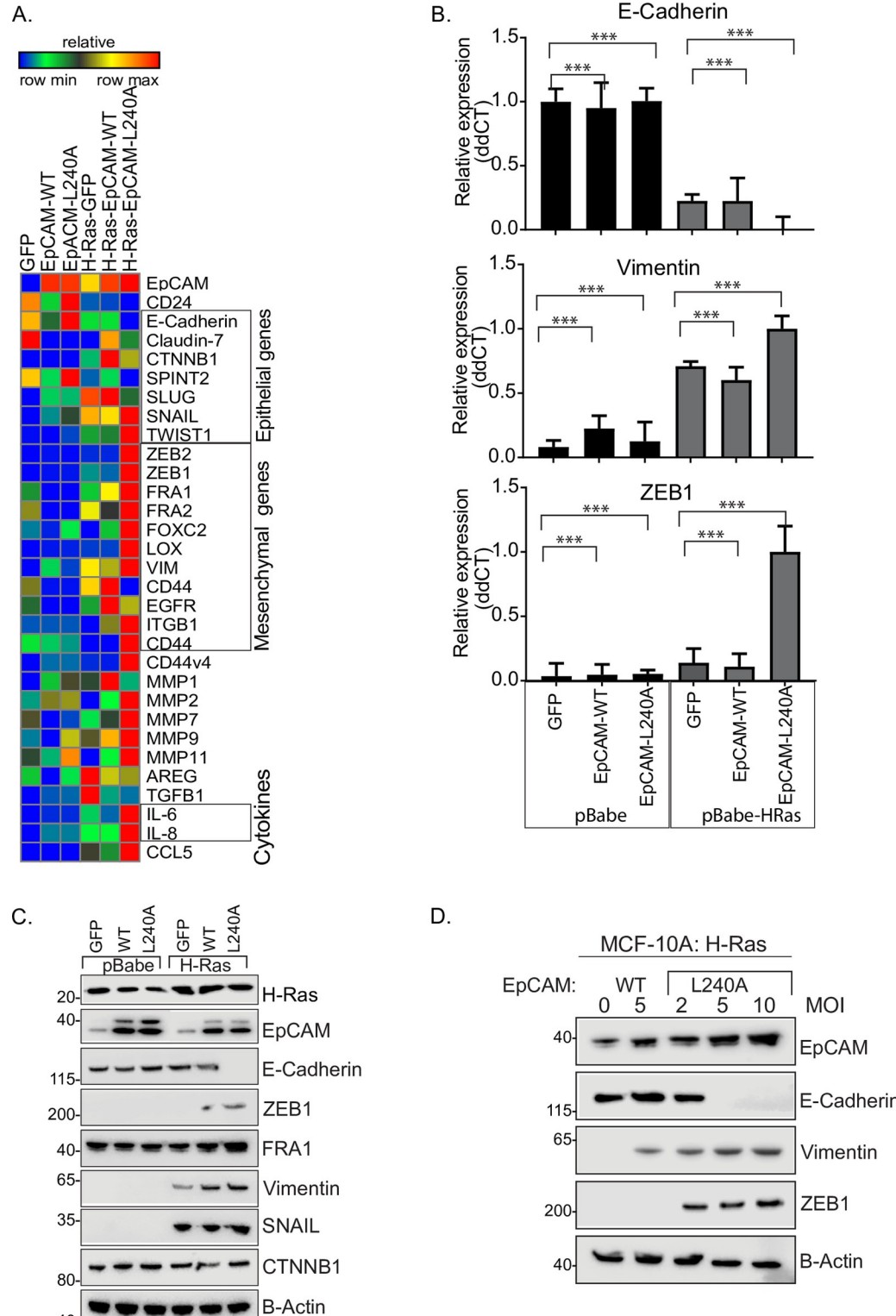

**Fig 4. EpCAM cooperates with Ras to induce EMT gene signature.** A. MCF10A cells transduced with control vector, Ras, EpCAM-WT and L240A were serum starved overnight and treated with growth media for 2 h. RNA extraction followed by cDNA synthesis and real-time PCR data is presented in a heat-plot. Results shows expression of epithelial, mesenchymal markers and cytokines. B. Quantitative relative ddCT data is presented for E-cadherin, vimentin and ZEB1. List of qRT-PCR primers used are listed in S1 Table. C. Cells were plated in 6CM plate and cultured for 48h followed by

lysate preparation and immunoblotting against epithelial marker, E-cadherin and mesenchymal markers ZEB1, FRA1, SNAIL and Vimentin. Results shows ZEB1, Snail and vimentin expressed in Ras-EpCAM-L240A cells. D. Ras-transduced cells were treated with EpCAM lentiviral particles EpCAM-WT (5 MOI) and EpCAM-L240A (2,5,10 MOI). Cell lysates were subjected to immunoblotting with EpCAM, E-cadherin, ZEB1, vimentin and actin antibodies. Results show that EpCAM-L240A induced ZEB1 expression in a dose-dependent manner.

are known to induce EMT in normal cells. ZEB1, a master regulator of EMT, was expressed together with vimentin (Fig 4B and 4C). This finding suggests that cytosolic EpCAM with H-Ras expression may have induced EMT through ZEB1 expression. To prove this, we generated EpCAM-WT and EpCAM-L240A lentiviral particles and added to H-Ras- transduced cells. As shown in Fig 4D, infection with 5 MOI of EpCAM-WT lentiviral particles did not affect E-cadherin expression; however, dose dependent treatments of 2, 5, and 10 MOI of EpCAM-L240A lentiviral particles induced ZEB1 and vimentin expression and suppressed E-cadherin expression. These results show that normal cells transduced with H-Ras together with cytosolic EpCAM-L240A can induce EMT through ZEB1. We validated this inverse expression correlation in a panel of cell lines with an EMT gene signature. Hierarchical cluster data analysis showed that the epithelial gene cluster is different from mesenchymal genes and EpCAM negative cells were positive for ZEB1 expression (S3 Fig). We have also shown that ZEB1 shRNA treatment induced EpCAM expression [9]. This result agrees with one independent study demonstrating that ZEB1 negatively correlated to 324 genes with EpCAM at the top of the list [22].

## Cytokine expression drives and maintains EMT

In our model system, Ras-EpCAM-L240A cells maintained an EMT phenotype without supplemental growth media. These results suggest that continued expression of cytokines may be inducing EMT through ZEB1. Inflammatory cytokines, including TGFβ, TNFα, IL-1, IL-6, and IL-8 activates transcription factors NF-κB, Smads, STAT3, Snail1/2, Twist1/2, and ZEB1/2, which drives EMT. Ras signaling can upregulate cytokines IL-1, IL-6, IL-8 in certain cell types [23]. We hypothesize that H-Ras-EpCAM-L240A cells produce growth factors and/or cytokines to maintain EMT. To investigate this possible phenomenon, conditioned media from cell cultures were assayed by an antibody-based cytokine array. Results showed that IL1α expression was restricted to H-Ras and H-Ras-EpCAM-WT cells; however, IL1β expression was observed in Ras-EpCAM-L240A cells (Fig 5A and 5C). These cells also exhibited high expression of proinflammatory cytokines IL-6 and IL-8 along with GM-CSF, ICAM-1, MIF, PAI-1 and CCL5 (Fig 5B and 5D). It is known that IL-6 promotes tumor growth. IL-8 exhibits tumor-promoting activity, including enhancing angiogenesis through paracrine signaling in Ras cells [24, 25]. Overall, H-Ras-EpCAM-L240A cells produced IL-6, IL1β, and GM-CSF, which are known to activate NF-kB, STAT3, and ERK signaling involved in EMT.

## EpCAM-transformed cells are sensitized to paclitaxel and quercetin

We sought to assess the efficacy of inhibiting common pathways that are likely activated in H-Ras-EpCAM-L240A cells. Transformed H-Ras-EpCAM-L240A cells were treated for 72h with most common signaling pathway inhibitors. Efficacy of signaling pathway inhibition was accessed immunoblotting phosphorylated and total proteins of ERK1/2, JNK, P38, AKT, NF-kB and Src (Fig 6A and 6B). MEK inhibitor (Ci-1040) treatment reversed cell morphology from a fibroblast-like to an epithelial-like morphology without significantly affecting cell death or proliferation (Fig 6A and 6C). Fig 6B–6D shows that treatment of cells with the JNK inhibitor SP600125 also did not cause significant changes in cell death or proliferation. The small

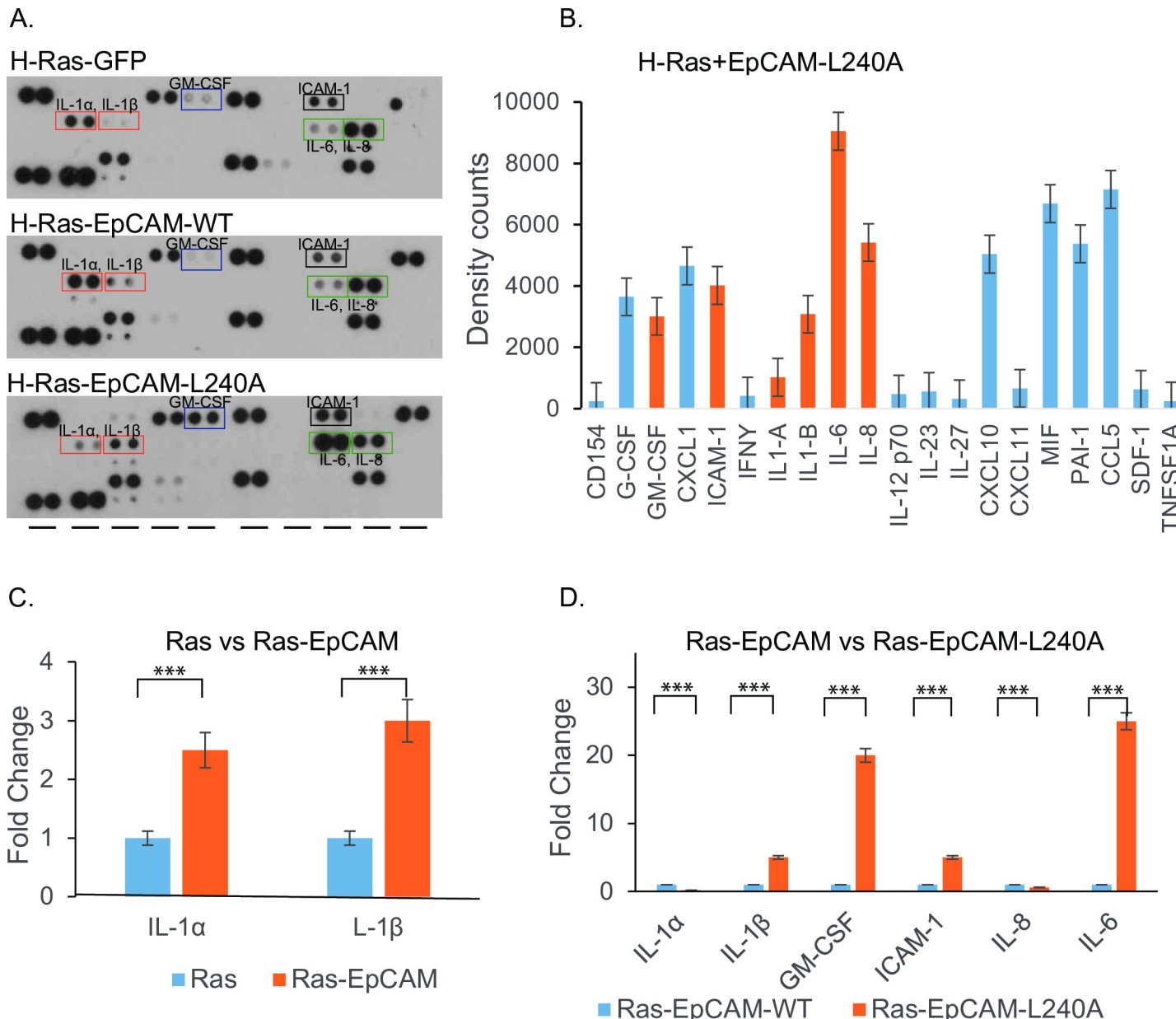

**Fig 5. Ras-EpCAM-induced EMT cells expressed inflammatory cytokines.** A-D. Ras-Pbabe vector, Ras-EpCAM-WT and Ras-EpCAM-L240A cells were cultured in Opti-Mem media (Thermo Fisher Scientific) for 48 h. Conditioned media was collected and applied to membrane-based antibody arrays (R&D Systems) and developed as recommended. Results show that IL1α and IL1β expression increased and was differentially expressed in EpCAM-WT and L240A cells compared to Ras cells. Ras-EpCAM-L240A cells show increased expression of GM-CSF, IL-6 and, IL1β.

molecular inhibitor for P38 MAPK (SB203580) and NF-kB-AKTi (BAY117082) induced about a 20% cell death rate. Dasatinib, an SRC family and Abl inhibitor, suppressed lamellipodia and filopodia formation necessary for cell migration. Doxorubicin and etoposide treatment did not alter significant cell death or proliferation. The most dramatic cell death (more than 90%) was observed after treatment with paclitaxel and the plant flavonoid quercetin. Paclitaxel is a chemotherapy drug for several cancers which targets microtubules. Quercetin is known to induce cell cycle arrest and apoptosis in some cancer cells; it prevents cell division, arrests cells in the

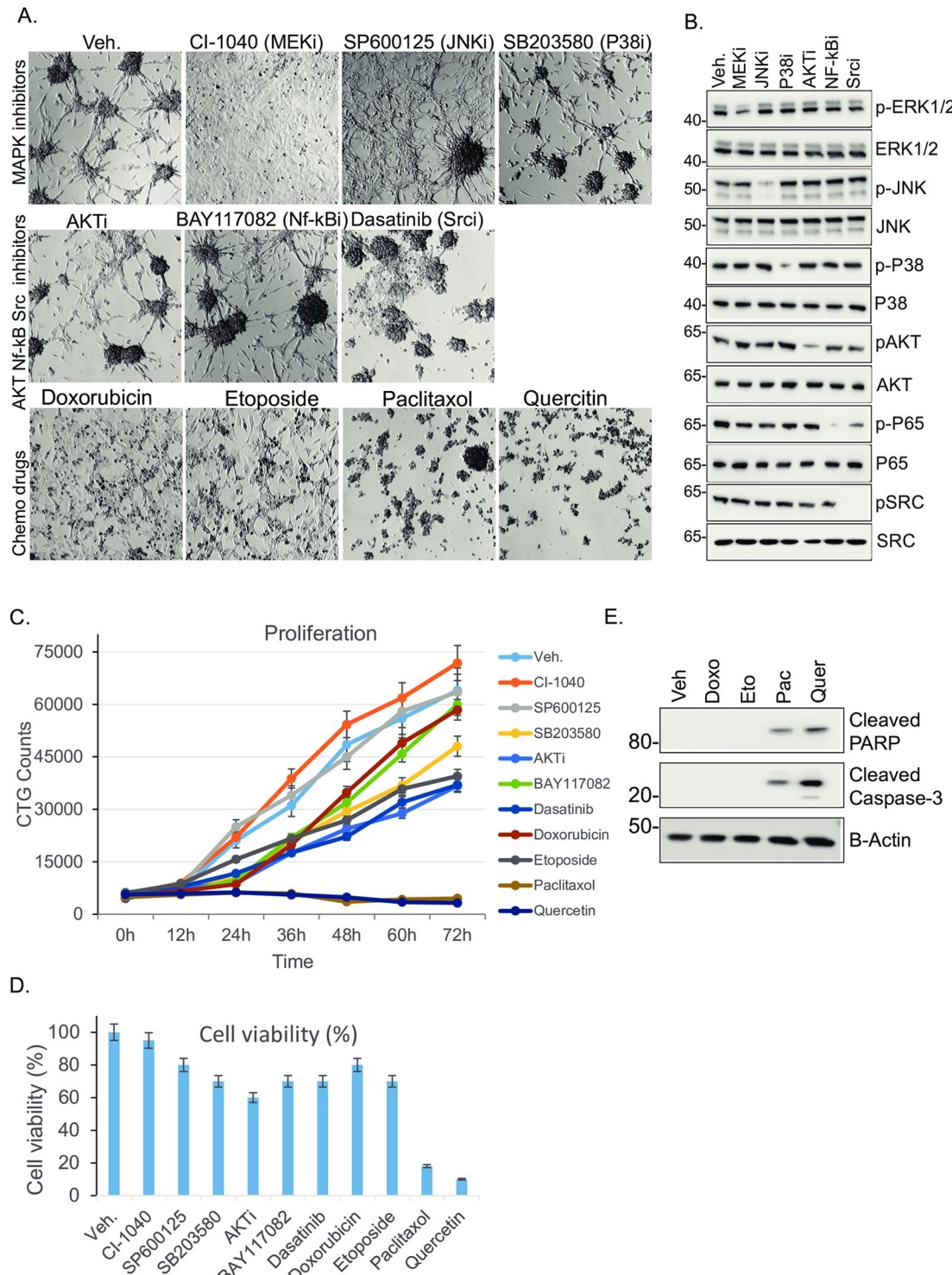

**Fig 6. Ras-transformed EpCAM cells are sensitized with paclitaxel and quercetin.** A-B. Phase contrast image of MCF10A-Ras-EpCAM-L240A cells cultured until sub-confluent and treated for 36-h with Veh. (DMSO), MEKi (Ci-1040, 20nm), JNKi (SP600125, 1 μM), P38 MAKPi (SB203580, 100 nM), AKti (LY294002, 2 μM), NF-kBi (BAY117082, 10 μM), dasatinib (100 nm), doxorubicin (2 μM), etoposide (5 μM), paclitaxol (100 nm), and quercetin (5 μM) to assay drug sensitivity. B. Efficacy of inhibitors used were analyzed by western blotting. Cells were harvested after 36h of drugs treatment, lysates were immunoblotted to common pathways, ERK1/2, JNK, P38, AKT, NF-kB and SRC. C. The cell proliferation assay was performed after 72 of plating using CellTiter-Glo Luminescent Cell Viability Assay. D. Cell viability was determined using MTT assay (see methods). Results shows paclitaxel and quercetin reduced cell growth significantly. E. MCF10A-Ras-EpCAM-L240A cells cultured until sub-confluent and treated for 36-h with Vehicle, Doxorubicin, Etoposide, Paclitaxel, and quercetin. Cell lysates were subjected for immunoblotting to assay cleaved caspase-3 and PARP.

G2/M-phase of the cell cycle, inhibits replication, and eventually causes apoptosis [26, 27]. In order to examine type of cell death induced by chemotherapy drugs, cells were treated with Doxorubicin, Etoposide, Paclitaxel and Quercetin for 72 hrs. We observed paclitaxel, quercetin affected cell proliferation within 24h of cell growth (Fig 6C). It is known that paclitaxel treatment results in cell death by expressing apoptosis-related proteins such as cleaved caspase-3 and cleaved PARP-1 [28]. Treated cells were harvested at 36h and analyzed for cleaved-PARP along with Caspase-3 (Fig 6E).Results demonstrated that paclitaxel and quercetin treated cell lysates detected cleaved caspase-3 and cleaved PARP. PARP is substrate for caspases, and caspase mediated cleavage of PARP is considered to be hallmark of apoptosis [29].

## Discussion

EpCAM plays an important role in developmental biology and the maintenance of intercellular adhesions. Knockout of EpCAM in mice can be embryonically lethal due to placental malformation and also the development of congenital tufting enteropathy [30]. Other knockout murine models demonstrate that EpCAM affects intercellular junction formation through aberrant recruitment and/or regulation of claudin proteins, E-cadherin, and β-catenin [31, 32]. In cancer, the prognostic significance of EpCAM expression is dependent on the tumor type. EpCAM overexpression is frequently associated with shorter patient survival in epithelial cancers [33]. In other cancer types, EpCAM expression is associated with improved outcomes. For example, high EpCAM expression in primary renal cell carcinoma tumors was associated with improved patient survival [34, 35], and in gastric cancer, loss of EpCAM expression was associated with aggressive tumor behavior [36]. In patients with esophageal squamous cell carcinoma, high EpCAM expression conferred a significantly higher survival rate [37]. EpCAM interacts with multiple signaling pathways, which can affect cancer growth and progression. EpCAM was reported to promote tumorigenesis by modulating PI3K/AKT/mTOR pathway activity [38, 39]. Protease-mediated cleavage of EpCAM generates EpICD fragments, which in turn activate Wnt-β-catenin signaling [40]. Our recent work shows that EpCAM expression suppresses ERK1/2 activation [9] and the protease cathepsin-L [5]. Both of these pathways are dominantly involved in promoting EMT, cancer cell invasion, migration, tumor metastasis, and drug resistance. We recently catalogued cancer associated EpCAM mutations and identified 115 unique cancer-associated somatic/missense mutations in the EpCAM coding region [5].

In the current study, we investigated the role of cell surface and cytosolic EpCAM expression in H-Ras-transduced epithelial cells. This work builds upon the insights of our prior study that examined the effects of cancer-associated EpCAM mutations on protein function [5]. Studies have revealed that overexpression of H-Ras is sufficient to initiate proliferation of cells in the absence of exogenous growth factors. H-Ras induces expression of growth factors HB-EGF, TGFα, and AREG [41, 42]. Our finding that cytosolic EpCAM (EpCAM-L240A) has the ability to further transform H-Ras-positive cells suggests cooperation between Ras and cytosolic EpCAM to promote tumorigenesis. The transformation of H-Ras-EpCAM-L240A cells into a mesenchymal cells (Figs 2 and 3) can be explained by at least 2 mechanisms. First, mutated EpCAM may generate endoplasmic reticulum stress due to protein misfolding. Since endoplasmic reticulum stress interferes with the normal physiological function of the cell, this may trigger survival pathway signaling and promote EMT. Secondly, bioinformatics analyses from scansite (https://scansite4.mit.edu/) predicts that EpCAM may be phosphorylated at serine, threonine, or tyrosine by cam and polo like kinases. Ras pathway signaling causes downstream protein phosphorylation and activates the mitogen-activated protein kinase (MAPK) cascade and the PI3K/AKT/mTOR pathway. Thus, cytosolic localization of EpCAM may increase the rate of EpCAM phosphorylation and alter EpCAM function. As shown in S2 Fig,

MCF-10A cells transduced with pBabe vector and EpCAM-WT maintained an epithelial phenotype. H-Ras- and EpCAM-L240A-transformed cells acquired mesenchymal characteristics. This observation is in agreement with a recent study of MCF10A-Ras cell transformation and altered morphology [43]. Matjaz et al. concluded that these morphological alterations can be mediated by several changes in epithelial architecture including: i) cell growth in two layers with varying E-cadherin expression and localization in the top layer and bottom layer, ii) decreased cellular adhesion due to loss of adhesion-like proteins, and iii) redistributed F-actin altering cellular tension. Based on this observation, we anticipate that cancer-associated EpCAM mutations that cause loss of membrane localization may a play significant role in contributing to cancer cell growth and EMT.

We have previously shown that the loss of EpCAM expression activated ERK1/2 signaling [9]. In this situation, it's highly possible that additional oncogene signaling as demonstrated here, can induce EMT through MAPKs. The loss of EpCAM through EMT [9], proteolytic cleavage [44], mutations [5], and aberrant glycosylation [45] are known to induce varied cellular phenotypes. Since EMT phenomena can be reversed to MET, it is important to understand mechanism involved in transitions. Our data suggested that EpCAM mutations together with Ras can transform cells, which suggests that EpCAM may play important role initiating EMT. EpCAM has 3 di-sulfide linkages which are generally present in secreted and membrane proteins for their role to stabilize the protein. EpCAM is also glycosylated at N74, N111, and N198 and dimerizes as an active molecule. Thyroglobulin type-I domains of EpCAM are conserved and are present in Trop2, p41 invariant chain, and insulin growth factor-like binding proteins. In these situations, structural changes or mis-folding through cancer-associated mutations may introduce detrimental role to designate as a driver mutations.

It is necessary to understand the cell signaling pathways modulated by EpCAM mutations in different cancer types. In the current study, we have shown Ras-EpCAM cells expressed inflammatory cytokines, IL-1α, IL-1β, IL-6, and IL-8 (Figs 4 and 5). These cytokines play crucial roles in tumor initiation, development, progression, and drug resistance. IL-8 expression contributes to cancer stem cell-like properties and poor prognosis in pancreatic cancer patients [46]. Ras-induced IL-8 expression plays a critical role in tumor growth and angiogenesis [25]. A potent cytokine, IL-6, plays an important role in lung cancer development [47]. Ras-EpCAM cells also expressed ZEB1. ZEB1 mediated IL-1β expression and promoted colon cancer cell stemness and invasiveness [48]. ZEB1 also served as a target gene of inflammatory cytokines [49]. Our findings here suggest that dysregulated mutant EpCAM expression may also generate stem cell phenotypes, since tumor-initiating cells are enriched with EpCAM [50, 51]. In line with these observations, the presence of EpCAM+/CD44+ cancer stem cells in tissue from colorectal cancer patients is significantly correlated with a more aggressive and higher grade tumor [52]. Only paclitaxel and quercetin treatment significantly affected cell proliferation and viability of MCF10A-Ras-EpCAM-240A cells (Fig 6). This suggests that tumors with EpCAM mutations can sensitize cancer cells to unique therapy regimens. EpCAM-based antibody therapies are still potential therapeutics; however, screening strategies based on EpCAM expression and localization are not available. Since, the Ras proteins have been termed "undruggable" [53], Ras-regulated genes such as EpCAM can serve as a better therapeutic target. Our data suggest that MEK inhibition reversed the mesenchymal morphology (Fig 6A). This is in agreement with reports in which MEK inhibitor monotherapy demonstrated modest efficacy due to transcription of negative feedback genes and the activation of PI3K-AKT survival pathway [54]. We anticipate that combined treatment strategy with survival pathway such as AKT may also induce cell death. EpCAM mutational data is being catalogued to a public TCGA database, and studying the role of these mutations in Ras- and RTK-driven tumorigenesis will help to develop future personalized therapies.

## Supporting information

**S1 Fig. A. Flow cytometry analysis of surfaced stained EpCAM**. MCF10-A cells transduced with Ras and EpCAM were analyzed using flow cytometry to distinguish between WT and cytosolic EpCAM. MCF-7 and T47-D cells were used as controls to monitor comparable expression of exogenous EpCAM. Cytosolic EpCAM is not detected in surface staining of cells. **B. EpCAM closely correlated genes are expressed together in Ras positive epithelial cells**. NCI-60 cancer cell lines (GSE5846) with Ras mutations were analyzed for expression of EpCAM and its "Nearest Neighbors" using GENE-E software.
(TIF)

**S2 Fig. Cell morphology.** MCF10A cells with vector, Ras, and EpCAM. Flat monolayer growth can be seen in MCF10-pBabe and EpCAM WT-cells. Protruding bi-layer like growth (see Discussion) can be seen in MCF10A-L240A and MCF10A-Ras-pBabe and EpCAM-WT cells. Totally altered morphology was observed in Ras-EpCAM-L240A cells.
(TIF)

**S3 Fig. EpCAM and ZEB1 expression.** Panel of breast cancer cell line (https://sites. broadinstitute.org/ccle/) were hierarchically analyzed using Gene-E software to separate epithelial and mesenchymal cell lines based on EMT gene signature. Left side rows designate mesenchymal cells and right side is for epithelial cells. EpCAM and ZEB1 expression can be seen in two different populations.
(TIF)

**S4 Fig. A. Full western blots**. Full western blots supporting Figs 1E and 4D. **B. Full western blots**. Full western blots supporting Fig 4C. **C. Full western blots**. Full western blots supporting Fig 6B. **D. Full western blots**. Full western blots supporting Fig 6E.
(TIF)

**S1 Table. List of qRT-PCR primers.** List of qRT-PCR primers.
(DOCX)

## Acknowledgments

We thank to Norton Thoracic Institute and St. Joseph's Hospital's Medical Center supporting this research. We thank Dr. T. Mohanakumar for reviewing and Billie Glasscock and Kristine Nally for assistance with editing and manuscript preparation.

## Author Contributions

**Conceptualization:** William E. Gillanders, Narendra V. Sankpal.

**Data curation:** Taylor C. Brown, Timothy P. Fleming.

**Formal analysis:** Timothy P. Fleming, Narendra V. Sankpal.

**Investigation:** William E. Gillanders, Ross M. Bremner, Narendra V. Sankpal.

**Methodology:** Fatma A. Omar, Narendra V. Sankpal.

**Project administration:** Narendra V. Sankpal.

**Resources:** Michael A. Smith, Ross M. Bremner.

**Supervision:** Narendra V. Sankpal.

**Visualization:** Narendra V. Sankpal.

**Writing – original draft:** William E. Gillanders, Narendra V. Sankpal.

**Writing – review & editing:** Taylor C. Brown, Timothy P. Fleming, Michael A. Smith, Narendra V. Sankpal.

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
