## [Decision Letter · Decision Letter 0]

20 Feb 2023

PONE-D-22-34859Cytosolic EpCAM cooperates with H-Ras to regulate epithelial to mesenchymal transition through ZEB1PLOS ONE

Dear Dr. Sankpal,

Thank you for submitting your manuscript to PLOS ONE. After careful consideration, we feel that it has merit but does not fully meet PLOS ONE’s publication criteria as it currently stands. Therefore, we invite you to submit a revised version of the manuscript that addresses the points raised during the review process.

We look forward to receiving your revised manuscript.

Kind regards,

Erika Di Zazzo

Academic Editor

PLOS ONE

Journal Requirements:

  "This research was supported by a startup grant from Norton Thoracic Institute, St. Joseph’s Hospital and Medical Center to NS. NO."

   "This research was supported by a startup grant from Norton Thoracic Institute, St. Joseph’s Hospital and Medical Center to NS. We thank Dr. T. Mohanakumar for reviewing and Billie Glasscock and Kristine Nally for assistance with editing and manuscript preparation."

  "This research was supported by a startup grant from Norton Thoracic Institute, St. Joseph’s Hospital and Medical Center to NS. NO."

Reviewers' comments:

Reviewer's Responses to Questions

**Comments to the Author**

1. Is the manuscript technically sound, and do the data support the conclusions?

Reviewer #1: Yes

Reviewer #2: Yes

2. Has the statistical analysis been performed appropriately and rigorously? 

Reviewer #1: No

Reviewer #2: Yes

3. Have the authors made all data underlying the findings in their manuscript fully available?

Reviewer #1: Yes

Reviewer #2: Yes

4. Is the manuscript presented in an intelligible fashion and written in standard English?

Reviewer #1: Yes

Reviewer #2: Yes

5. Review Comments to the Author

Reviewer #1: A western blot analysis of EpCAM is suggested to implement figure 1

Indicate the time of experiments shown in figure 2.

Statistical analysis is lacking in all the panel figures.

Proliferation assays at different time points is suggested and a different representation (curve instead of histograms)

Add a western blot analysis of all EMT markers tested in all experiments.

To Assess the efficacy of inhibitors on several signalling pathways a western blot analysis should be included.

Reviewer #2: In this manuscript, the authors investigated the role of cytosolic EpCAM in MCF-10A cell line modified to overexpress the H-Ras oncogene. The manuscript is well written and well structured, with several experiments aimed at defining the interconnected role of H-Ras overexpression and EpCAM in EMT. Figures and graphs are well defined.

There are only a few comments to further improve the work and make it even more complete.

Point 1: The role and possible collateral effects that targeted inhibition of EpCAM would have in epithelial tumors should be expanded.

Point 2: Did you analyze and define the type of cell death involved in the treatment with Paclitaxel and Quercetin? If yes, it would be interesting to add it in the results.

Point 3: In the materials and methods section please add more information about the inhibitors used to test the EpCAM-transformed cells.

Point 4: In the materials and methods section there are some subsections in which the number of cells seeded to perform the experiments has not been included. Please add them.

Point 5: Again, in the materials and methods section, expand the statistical analysis part.

6. PLOS authors have the option to publish the peer review history of their article (what does this mean?). If published, this will include your full peer review and any attached files.

Reviewer #1: No

Reviewer #2: No

---

## [Author Response · Author response to Decision Letter 0]

8 Apr 2023

Dear Editor and Reviewer,

We are grateful to you for reviewing our manuscript (Title: Cytosolic EpCAM cooperates with H-Ras to regulate epithelial to mesenchymal transition through ZEB1, PONE-D-22-34859). 

Your comments on this revision were very valuable and helpful for improving the quality of this manuscript. Reviewer’s comments are laid out below in italicized font. Our responses are given in normal font.

Review Comments to the Author

Reviewer #1: 

A western blot analysis of EpCAM is suggested to implement figure 1

Response: Thank you for the an important suggestion, we have added western blot data for EpCAM together with Ras and loading control Beta actin as a new Figure 1E. 

Indicate the time of experiments shown in figure 2.

Response: Figure 2A, B, C and D experimental times are noted in Figure 2 legends and result section (Page 12). 

Statistical analysis is lacking in all the panel figures.

Response: Thank you for raising important point. We have re-analyzed the data and statistical analysis is done on all figures, new figures are presented and analysis part mentioned in method section. 

Proliferation assays at different time points is suggested and a different representation (curve instead of histograms)

Response: Thank you for very valid suggestion. Two proliferation assays at Figure 2A and Figure 6C are changed to different time points and plotted as curve. 

Add a western blot analysis of all EMT markers tested in all experiments.

Response: Thank you for bringing this up, to support our Figure 4A and B, we have added western blot panel as Figure 4C we have added, Ras, EpCAM and related EMT markers (E-cadherin, Zeb1, FRA1, VImentin, Snil and CTNNB1) as suggested. 

To assess the efficacy of inhibitors on several signaling pathways a western blot analysis should be included.

Response: Thank you for suggestion. We have ran western blot analysis as suggested for all the signaling pathway inhibitors used in the studies including ERK, JNK,P38,AKT, NF-kB, and Src. Phosphorylated/Activated and the total proteins are loaded and shown in new figure 6B.

Reviewer #2: In this manuscript, the authors investigated the role of cytosolic EpCAM in MCF-10A cell line modified to overexpress the H-Ras oncogene. The manuscript is well written and well structured, with several experiments aimed at defining the interconnected role of H-Ras overexpression and EpCAM in EMT. Figures and graphs are well defined. There are only a few comments to further improve the work and make it even more complete.

Point 1: The role and possible collateral effects that targeted inhibition of EpCAM would have in epithelial tumors should be expanded.

Response: Thank you for suggesting important part. We are expanding the study in lung and other cancers where we have seen up to 5% mutations in EpCAM gene. Inhibition of EpCAM can be achieved by antibody, shRNA or CRISPR. The current study used normal cell line. The limitation in the current study is that, we already have transduced cells 2 times with Ras and EpCAM, additional viral transductions generates stress affecting cell growth and overall cell morphology.

Point 2: Did you analyze and define the type of cell death involved in the treatment with Paclitaxel and Quercetin? If yes, it would be interesting to add it in the results.

Response: We understand your valid point. Our initial goal was to seek and target the cell death. However, we are still working the major apoptosis mechanism in different mutation. Our goal of this study was to demonstrate that EpCAM mutations can be targeted successfully which we have presented. However, we have added a separate figure panel showing caspase-3 and PARP as the major contributor. The related text has been added on page 16 in result section. 

Point 3: In the materials and methods section please add more information about the inhibitors used to test the EpCAM-transformed cells.

Response: Thank you noticing our missed point from our side. We have added all inhibitors and their doses used the studies in methods section, page and Figure 6 legends.

Point 4: In the materials and methods section there are some subsections in which the number of cells seeded to perform the experiments has not been included. Please add them.

Response: Thank you for pointing the missing part, in 6 different subsections. We have added number of cells in each section of methods. 

Point 5: Again, in the materials and methods section, expand the statistical analysis part.

Response: As suggested we have expanded the statistical analysis part in method section and revised all the related figures.

---

## [Editor Report · Decision Letter 1]

2 May 2023

Cytosolic EpCAM cooperates with H-Ras to regulate epithelial to mesenchymal transition through ZEB1

PONE-D-22-34859R1

Dear Dr. Sankpal,

We’re pleased to inform you that your manuscript has been judged scientifically suitable for publication and will be formally accepted for publication once it meets all outstanding technical requirements.

Kind regards,

Erika Di Zazzo

Academic Editor

PLOS ONE